# The Relationship between Food Healthiness, Trust, and the Intention to Reuse Food Delivery Apps: The Moderating Role of Eco-Friendly Packaging

**DOI:** 10.3390/foods13060890

**Published:** 2024-03-15

**Authors:** Kyung-A Sun, Joonho Moon

**Affiliations:** 1Department of Tourism Management, Gachon University, Seongnam 13120, Republic of Korea; kasun@gachon.ac.kr; 2Department of Tourism Administration, Kangwon National University, Chuncheon 24341, Republic of Korea

**Keywords:** food delivery app, food healthiness, trust, intention to reuse, eco-friendly food packaging

## Abstract

The goal of this research is to investigate the relationship among food healthiness, trust, and the intention to reuse food delivery apps. Another purpose of this work is to examine the moderating effect of eco-friendly food packaging on the association between food healthiness and trust in food delivery apps. A survey was the main instrument for this work, with Amazon Mechanical Turk being used to collect the relevant data, resulting in a total of 343 observations. PROCESS model 7 was employed to test the research hypotheses. The results reveal that the intention to reuse is positively impacted by trust and food healthiness in food delivery apps. The results also uncover a significant moderating impact of eco-friendly packaging on the relationship between food healthiness and trust. The high food healthiness and high eco-friendly packaging group has the highest level of trust, while the low food healthiness and low eco-friendly packaging group has the lowest. The results of this research are therefore important because they clarify the relationship among these four attributes. Moreover, the results of this study have notable managerial implications.

## 1. Introduction

There is fierce competition in the market of food delivery application systems. Statista [1] has reported that multiple companies are competing in this market, with Door Dash possessing the largest market share (65 percent), while Uber Eats (23 percent) and Grubhub (9 percent) rank second and third, respectively. In this context, food delivery application service businesses must allot their resources more adequately to survive. The first step in resource allocation could involve understanding market characteristics. Thus, this study explores the characteristics of food delivery application users.

The first part of this work concerns the intention to reuse. The intention to reuse is a popular attribute in research because the repetitive use of certain services or systems results in increased market share. In fact, numerous scholars have examined user characteristics by using the intention to reuse as the dependent variable in the context of online transactions [2,3,4]. Given the popularity of these apps, the authors of this research selected the intention to reuse as the dependent variable in order to understand the characteristics of food delivery application system users. The next focus in this study is food healthiness. Economic development has caused better living conditions; individuals have therefore become more interested in their health conditions [5,6]. To follow this market trend, food sellers offer food that promotes health conditions [7,8]. However, scholars have noted that the weakness of products offered via food delivery application services lies in health-related issues, as deliverable food has constraints in nutrition quality [9,10,11]. Therefore, it is worthwhile to explore the characteristics of food healthiness regarding how users of food delivery application systems perceive that healthiness. Accordingly, this research investigates the role of food healthiness as an independent variable.

The third element in this research is trust, which concerns the credibility of a seller according to the consumer behavior literature [12,13,14]. Many studies have demonstrated the role of trust as a mediator. This suggests that trust could impact the domain of food delivery application services [15,16,17]. Thus, trust is selected as the mediator in this study. The final domain in this work is eco-friendly packaging. Nguyen et al. [18] have argued that the effect of eco-friendly packaging perception is unclear; hence, it is worthwhile to reveal the impact of eco-friendly packaging, from the perspective of consumers, because this is useful in terms of business information. Prior studies have also stated that environmental perception could be a moderating variable for understanding consumer characteristics [19,20]. Hence, because packaging is an essential component of delivery services, understanding consumer perceptions of eco-friendly packaging is important. However, scholars have rarely explored the characteristics of eco-friendly packaging among food delivery application system users. Thus, this research scrutinizes the moderating effect of eco-friendly packaging in the food delivery application system context. 

Overall, the purpose of this research is to examine the relationship among food healthiness, eco-friendly packaging, trust, and the intention to reuse. This research thus theoretically contributes to the literature by clarifying the relationship among these four attributes. In order to accomplish the research purpose, this work employed a survey using confirmatory factor analysis and PROCESS macro model 7 to test the validity of mesaurement items and for hypotheses testing, respectively. This work sheds light on the literature by disclosing consumer perceptions to food delivery applications’ services, with a focus on eco-friendly packaging and food healthiness. This is because not only is food healthiness likely to become a weakness of the food delivery application business mode but also because it is unclear whether eco-friendly packaging might build a competitive advantage for a business. Moreover, the results of this work provide insights into the management of food delivery application system services in terms of how to handle food healthiness and eco-friendly packaging in business.

## 2. Literature Review and Hypothesis Development

### 2.1. Food Healthiness

Prior studies have defined food healthiness as consumers’ perceptions of whether food in the market promotes health conditions [5,6,21]. The market has become more interested in health conditions because poor health conditions cause enormous costs, including medical costs, and are obstacles to economic and social activity [22,23,24]. Regarding this aspect, the food business has focused more on offering healthy food because consumers are willing to pay for it [5,7,8]. Many scholars have also examined food healthiness in the food market sector. Ali and Ali [7] have studied the characteristics of food healthiness, and Chan and Zhang [6] have examined the link among food healthiness, consumer perception, and consumption. Topolska et al. [8] also contend that food healthiness encourages consumers to make a purchase decision. Tuorila and Hartmann [25] imply that markets value greater food healthiness, an avenue toward business sustainability. Moreover, Samoggia and Riedel [26] state that services providing nutrition values for healthier consumption have gained more popularity in the market. In the literature review of Plasek et al. [27], the authors suggest that consumers are interested in consuming food that promotes their health conditions. Chan and Zhang [6] have also found that food healthiness is an imperative attribute for consumer decision-making. Accordingly, the above review of the literature implies that the market has placed increasing value on food healthiness and that, overall, food healthiness represents a significant trend in the food market.

### 2.2. Eco-Friendly Packaging

Eco-friendly packaging refers to whether a packaging material is harmful to the environment [28,29,30]. La Torre et al. [31] have stated that whether a packaging material is eco-friendly is related to whether the chemical substance it is made of undermines individual health conditions. Global warming has led consumers to focus more on eco-friendly packaging because their concerns over the environment have been increasing over time [32,33,34]. Moreover, Nguyen et al. [18] claim that consumers have become more interested in eco-friendly packaging because plastic packaging materials contain microplastics and toxic materials that impair individual health conditions. Moustafa et al. [35] and Marrez et al. [36] also note that eco-friendly food packaging provides a perception of safety to consumers in the food business sector. In the context of food delivery app services, Shimul and Cheah [37] argue that pride and guilt are the critical psychological mechanisms for choosing eco-friendly packaging among consumers. Chen and Lee [38] also document the importance of environmental packaging issues. Plasek et al. [27] have shown that food packaging is a critical attribute for consumer decision-making because it is correlated with food quality. Ketelsen et al. [39] have stated that consumers understand that food packaging causes environmental problems but that their actual awareness of food packaging is trivial. Additionally, scholars in the domain of food delivery application services have emphasized the environmental aspect of persuading consumers [38,40,41]. However, few studies have explored the impact of eco-friendly packaging on consumer evaluation.

### 2.3. Trust

Scholars have stated that trust is a reliable perception of a subject [12,13,14]. Jones et al. [13] have claimed that trust plays a significant role in decision-making in the electronic commerce sector. Moreover, Bianchi and Andrews [42] and Yoon and Occena [43] have investigated the determinants of trust by examining users of electronic commerce. Among previous studies, many works have explored both the antecedents and consequences of trust in the online transaction area. This suggests that the role of trust is as a mediator. In fact, numerous scholars have demonstrated the role of trust as a mediator in the online transaction system domain [15,16,17]. Specifically, Chang and Chen [16] have unveiled the role of trust as a mediator among online store users. Ibrahim et al. [44] have revealed that trust is a mediator between social media marketing and loyalty behavior in the online marketing of café businesses. Nguyen and Pervan [45] have also found that consumer trust is an imperative aspect by which to assess consumer perception of corporate social responsibility. Furthermore, Macready et al. [46] have shown that trust is a crucial attribute for investigating consumer perception in the food value chain.

### 2.4. Intention to Reuse

The intention to reuse refers to the degree of a user’s desire to adopt a certain technological instrument again [47,48,49]. The intention to reuse is a kind of loyalty behavior because more frequent use of a certain technology brings about more sales [2,50]. Indeed, many studies have explored the determinants of the intention to reuse. Li et al. [48] have examined the attributes that influence the intention to reuse e-learning systems, and Arahita and Hatammimi [2] have documented the determinants of the intention to reuse in the domain of mobile banking services. Moreover, Daassi and Debbabi [50] have researched the characteristics of augmented reality application system users, using the intention to reuse as their dependent variable. Hussein et al. [4] have also studied the antecedents of the intention to reuse Google Classroom, and Mattila et al. [51] have revealed the factors affecting the intention to reuse in the area of mobile car sharing services. Additionally, Lee et al. [52] have examined customers of low-cost carrier businesses, using the intention to reuse as their dependent variable. Concerning food delivery apps, Cho et al. [53], Alalwan [54], and Bao and Zhu [3] have disclosed the determinants of the intention to reuse. According to the above review of the literature, the intention to reuse has therefore been widely explored by numerous scholars.

### 2.5. Hypothesis Development

Previous research has argued that selling food that can improve health conditions builds credible perceptions among customers [55,56]. Specifically, Ward et al. [57] have found that healthy and nutritious food is an essential element for the building of consumer trust among south Australian consumers. Macready et al. [46] have researched five European countries, demonstrating that consumer trust is significantly associated with food healthiness. Moreover, Konuk [58] has shown a similarly positive impact of food healthiness on trust. The findings of a literature review by Wu et al. [59] also indicate that the offering of healthier food plays a significant role in building consumer trust. This study thus hypothesizes the following:

**Hypothesis** **1.**
*Food healthiness exerts a positive effect on trust among food delivery app users.*


The next domain is that of the moderating effect of eco-friendly packaging. Previous research has contended that eco-friendly packaging functions as a clue for product appraisal [60,61]. Though it is not sufficient for appraising certain products precisely, packaging enables consumers to estimate their quality [20,61]. Nguyen et al. [18] also argue that eco-friendly packaging leads to more attractive perceptions among consumers. Kim et al. [20] and Ebrahimi et al. [19] have additionally documented the way in which an environmental appeal improves the perception of health-related products; both studies also reveal the moderating effect of environmental aspects in the consumer behavior domain. Additionally, Popovic et al. [62] have carried out a systematic review of the literature and found that packaging and eco-friendliness are more imperative for consumers because food packaging is related to food safety. Nguyen et al. [63] have uncovered how consumer appraisal of food is improved by eco-friendly packaging because environmental packaging demonstrates a perception of improved safety for the food it contains. The above literature review therefore shows that food delivered in eco-friendly packaging enhances customers’ assessments. Eco-friendly packaging is likely to make healthier food seem safer, which leads consumers to trust this food more. Therefore, this research proposes the following hypothesis:

**Hypothesis** **2.**
*Eco-friendly packaging significantly moderates the association between food healthiness and trust among food delivery app users.*


The third area in this work is that of the relationship between healthiness and the intention to reuse. The findings of d’Astous and Labrecque [61] indicate that food healthiness is an imperative element in consumer decision-making. Moreover, Kim et al. [64] have revealed that revisit intentions towards restaurants are positively influenced by food healthiness. Additionally, Ji et al. [65] have found that food healthiness plays a significant role in building the intention to revisit among café customers. Suhartanto et al. [66] have explored the users of food delivery application systems and found that food healthiness exerts a positive impact on loyalty. Francioni et al. [67] have also examined food delivery application users, and their results show a positive and significant effect of food healthiness on reuse intention. Based on the above review of the literature, this study thus proposes the following:

**Hypothesis** **3.**
*Food healthiness exerts a positive effect on the intention to reuse among food delivery app users.*


Prior studies have shown that trust plays a significant role in building customer loyalty behavior, such as the intention to reuse and repurchase, because vendor credibility is not easy to establish [68,69]. Regarding the outcomes of empirical studies, Prayudi et al. [70] have demonstrated the positive effect of trust on the intention to reuse among mobile banking service users. In addition, Malhotra et al. [71] have disclosed the positive impact of trust on the intention to reuse among online shoppers. Furthermore, scholars have revealed that trust positively affects the loyalty of users in several online service business areas [72,73,74]. Based on the above literature review, this study therefore hypothesizes the following:

**Hypothesis** **4.**
*Trust exerts a positive effect on the intention to reuse among food delivery app users.*


## 3. Method

### 3.1. Research Model and Data Collection

Figure 1 illustrates the research model. Healthiness is the explanatory variable, and trust is the mediator. Healthiness exerts a positive effect on trust. The intention to reuse is the dependent variable in this work. The intention to reuse is positively influenced by both healthiness and trust. Additionally, this research used eco-friendly packaging as the moderating variable in the relationship between healthiness and trust.

Amazon Mechanical Turk was used to collect data for this research. Amazon Mechanical Turk has been widely employed by researchers to collect survey responses. Additionally, Amazon Mechanical Turk has fewer time and geographical constraints when collecting data because it uses an online panel, though it offers less technical proficiency and less sincere responses than a face-to-face survey method might [21,75,76]. Respondents voluntarily participated in the survey by accessing the Amazon Mechanical Turk system as a worker for financial reward. In fact, many studies have documented that the quality of the data from Amazon Mechanical Turk is sound for statistical inference [75,76,77]. Given the credibility, this research selected Amazon Mechanical Turk as an instrument of data collection. Moreover, this research mainly targeted the American food delivery application service market due to its enormous market size [1]. Additionally, because food delivery apps users require minimum skill in order to handle the online system, using Amazon Mechanical Turk is likely to become an appropriate selection. Amazon Mechanical Turk is also an adequate instrument for data collection because its panels mostly include Americans, allowing researchers to focus primarily on American consumers [78]. Data collection for this research was implemented between 1 and 6 August in 2023. There were initially 343 observations for data analysis. By establishing a survey system that uses a required option for survey participants, this study aimed to minimize the problems caused by unanswered questions. Thus, this study used a total of 343 observations for the data analysis.

Table 1 shows the profile of the survey participants. There was a total of 343 observations and 208 and 135 male and female participants, respectively. Table 1 presents their ages. Approximately 60 percent of participants were younger than 40 years old, and most of the participants were employed. Table 1 also illustrates the information on monthly household income and weekly use-frequency of food delivery apps.

### 3.2. Measurement Items

Table 2 shows the measurement items. There are four constructs: healthiness, eco-friendly packaging, trust, and the intention to reuse. All four constructs are measured by four items. This research used a 5-point Likert scale (1: strongly disagree, 5: strongly agree) to measure healthiness, eco-friendly packaging, and the intention to reuse. Additionally, this research used a semantically different 5-point scale (1: untrustworthy, 5: trustworthy) to measure trust. Regarding definition, food healthiness represents the way in which consumers perceive food in order to promote health conditions in the domain of food delivery apps [5,6,7]. Eco-friendly packaging is defined as how consumers assess how packaging of food delivery apps protects the natural environment [28,30,37]. The definition of trust refers to the way in which consumers have credible perceptions of food delivery apps [15,17,44]. Finally, the definition of the intention to reuse is the degree of consumer intention to reuse food delivery apps [3,48,49].

### 3.3. Data Analysis

In this work, frequency analysis was implemented in order to obtain the information of survey participants. Confirmatory factor analysis was performed to ensure convergent validity; the criteria of convergent validity include loading > 0.5, average value extracted (AVE) > 0.5, and construct reliability (CR) > 0.7 because this work adopted multiple items for the measurement of variables in the survey [79,80,81]. The goodness of fit for confirmatory factor analysis was confirmed by the following criteria: root mean square residual (RMR) < 0.05, goodness of fit index (GFI), normed fit index (NFI), relative fit index (RFI), incremental fit index (IFI), Tucker–Lewis index (TLI), and comparative fit index (CFI) > 0.8 [79,80,81]. Then, this study calculated mean values and standard deviations for the constructs. The correlation matrix was employed to examine the relationship. This study used the PROCESS macro with statistical packaging in social science (SPSS) 20.0 version. PROCESS model 7 was employed. PROCESS model 7 is based on an ordinary least squares regression for the path analytic model to analyze not only the conditional indirect effects of the mediated moderation model but also the effect of moderating effect based on varied magnitude using moderating variable and its significance and change of R^2^ [82]. The model produced bias-corrected 95 percent bootstrap confidence intervals for conditional indirect effects based on 5000 bootstrap samples [82]. The interaction variable was computed to examine the moderating effect (healthiness × eco-friendly packaging). Additionally, median split analysis was conducted in this research to examine the moderating effect further by calculating the mean value for the four groups. The median values of both healthiness and eco-friendly packaging are 4. Based on the results of the median split analysis, this research used four groups: (1) high eco-friendly packaging and high food healthiness group, (2) high eco-friendly packaging and low food healthiness group, (3) low eco-friendly packaging and high food healthiness group, and (4) low eco-friendly packaging and low food healthiness group.

## 4. Results

### 4.1. Results of Confirmatory Factor Analysis and Correlation Matrix

Table 3 exhibits the results of the confirmatory factor analysis. All values of factor loading are greater than the cutoff value of 0.5. All CR and AVE values are greater than the threshold CR (0.7) and AVE (0.5). The results indicate that the convergent validity of the measurement items was acceptable. The goodness of fit indices suggest that the results are acceptable (RMR = 0.038, GFI = 0.844, NFI = 0.894, RFI = 0.870, IFI = 0.917, TLI = 0.898, and CFI = 0.916). In addition, Table 3 illustrates the mean and standard deviation of the following variables: food healthiness, eco-friendly packaging, trust, and the intention to reuse.

Table 4 shows the correlation matrix. Healthiness positively correlates with eco-friendly packaging (r = 0.918, *p* < 0.05), trust (r = 0.662, *p* < 0.05), and the intention to reuse (r = 0.663, *p* < 0.05). Eco-friendly packaging also positively correlates with trust (r = 0.678, *p* < 0.05) and the intention to reuse (r = 0.660, *p* < 0.05). Moreover, trust positively correlates with intention to reuse (r = 0.772, *p* < 0.05).

### 4.2. Results of Hypothesis Testing

Table 5 shows the results of hypothesis testing by Hayes’ PROCESS model 7. Both models are statistically significant given the F values (*p* < 0.05). The results show that healthiness × eco-friendly packaging exerted a positive impact on trust (β = 0.177, *p* < 0.05). This indicates the significant moderating effect of eco-friendly packaging on the relationship between healthiness and trust. Moreover, healthiness positively affects the intention to reuse (β = 0.287, *p* < 0.05), and trust positively impacts the intention to reuse (β = 0.538, *p* < 0.05). Regarding the index of mediated moderation (index: 0.0956, *p* < 0.05), the effect appeared to be significant. Regarding the R^2^, the explanatory attributes accounted for the 43.54 percent of variability for trust. Additionally, R^2^ in model 2 indicates the 48.57 percent of explanatory power of the explanatory variables for intention to reuse.

Figure 2 shows the results of the median split analysis used to scrutinize the moderating effect of eco-friendly packaging on the relationship between healthiness and trust. The high eco-friendly packaging and high healthiness group showed the highest mean value (mean = 4.498), whereas the mean of the low eco-friendly packaging and low healthiness group was the lowest (mean = 3.660). 

## 5. Discussion

This study has explored the relationship among food delivery app food healthiness, eco-friendly packaging, trust, and the intention to revisit. Regarding the mean values, trust presented the highest value (mean = 4.126), while eco-friendly packaging showed the lowest mean value (mean = 3.802). Hence, consumers maintain a reliable perception of the food delivery app service when shopping but remain somewhat skeptical of the environmental aspect of any food delivery app service. According to the results of the hypothesis testing, trust also positively affects the intention to reuse food delivery apps. The results are aligned with the findings of prior studies, i.e., that trust plays a significant role in increasing the sales of businesses [70,73,74]. Moreover, the results have revealed that consumers with positive perceptions of their food’s healthiness are more likely to use food delivery apps again. In a similar vein, Suhartanto et al. [66] have found that food healthiness is a critical attribute for building more consumer loyalty among food delivery apps. Regarding the moderating effect of eco-friendly packaging, consumers might value eco-friendlier packaging when consuming healthier food products because micro plastics and the effects of food temperature on plastic are likely to exert negative effects on their health conditions [18,37,83]. The healthy perception of a food might include its packaging because ideas of food conditions related to health might be influenced in this way [18,84]. Therefore, the results of this work indicate that offering food with bioplastic containers might elevate consumer trust in the domain of food delivery apps. However, the association between trust and healthiness appears to be converse to the proposed research hypotheses. This might be explained by an argument in prior studies that suggests that food delivery app products could be deemed the culprits of individuals’ obesity and negative health conditions because most menus in food delivery apps contain high calories, sugar, and fat [9,10,11]. It is implied that most menus in food delivery app services contain fast food that are not guaranteed to be sanitary. Moreover, this research is valuable, in terms of external validity, for healthiness, trust, and the intention to reuse. Specifically, this research affirms the findings in the literature for the effect of trust [68,69] and food healthiness [61,64] on the intention to reuse, which could become the external validity of this work.

## 6. Conclusions

The results have academic implications. This study extends the literature by demonstrating the moderating impact of environmental packaging on the relationship between food healthiness and trust in food delivery apps. Although packaging is an essential issue in the food delivery app domain, its effect on consumer perception has been insufficiently explored. Aiming to minimize this research gap, this research has posited and demonstrated the moderating effect of food packaging. 

This work also has practical implications. First, food delivery app service managers might focus more on elevating consumers’ trust. This could be accomplished by dedicating their resources to the public welfare sector, such as environmental, social and governance (ESG) implementation. However, careless ESG might hinder the effective resource allocation of businesses. Thus, food delivery app business managers might focus more on imperative stakeholders with their business model. Moreover, food delivery app business managers might consider the healthy food sector in their menu composition because such an option might lead consumers to use their food delivery app again. Additionally, the negative effect of healthiness on trust is likely to become an opportunity for food delivery app service providers because the healthy food market is likely to be a potential blue ocean. Additionally, food delivery app business managers might contemplate eco-friendly packaging options because consumers might perceive these as healthier or safer in their food consumption. Offering both healthier and more environmentally friendly options might be part of ESG, which might positively impact consumer trust because it is likely to lead consumers of a food delivery app to gain a higher perception of trust through eco-friendliness and by considering the health conditions of its consumers.

This study has certain limitations. First, the survey participants of this research were limited to the online panel. Because of their familiarity with technology and their general proficiency to control technology the survey participants were relatively young. This might undermine the generalizability of this work. Future research might consider either offline surveys or experimental designs for more vivid outcomes, integrating a wider range of participant ages. Moreover, the moderating variable in this research was only eco-friendly packaging, which might be a limitation. If future studies contemplate various moderating attributes, it might allow them to identify more food delivery app users. This research only concentrated on the environmental aspect of packaging, though packaging has other aspects, such as aesthetics and convenience. Researchers might be able to consider other attributes for packaging and the results might become more significant in understanding the behavior of food delivery app users.

## Figures and Tables

**Figure 1 foods-13-00890-f001:**
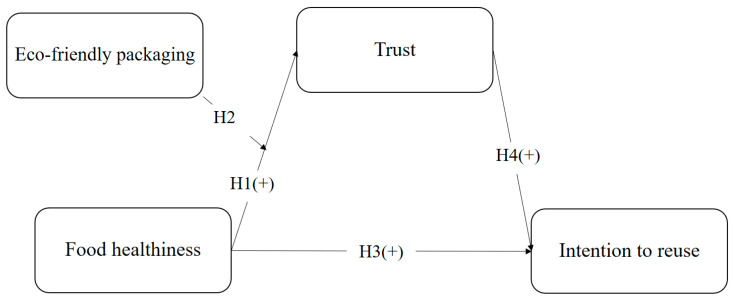
Research model.

**Figure 2 foods-13-00890-f002:**
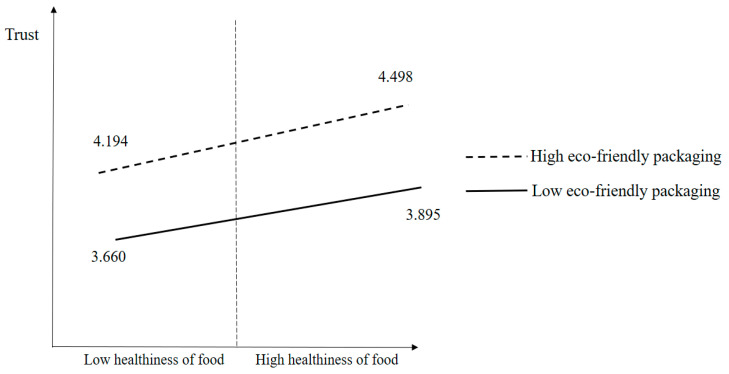
Results of the moderating effect of eco-friendly packaging.

**Table 1 foods-13-00890-t001:** Profiles of survey participants (N = 343).

Item	Frequency	Percentage
Male	208	60.6
Female	135	39.4

20 s	102	29.7
30 s	137	39.9
40 s	72	21.0
50 s	15	4.4
Older than 60	17	5.0

Unemployed	7	2.0
Employed	336	98.0

Monthly household income		
Less than $2000	34	9.9
$2000~$3999	52	15.2
$4000~$5999	93	27.1
$6000~$7999	64	18.7
$8000~$9999	47	13.7
More than $10,000	53	15.5

Weekly application using frequency		
Less than 1 time	46	13.4
1~2 times	148	43.1
3~5 times	125	36.4
More than 5 times	24	7.0

**Table 2 foods-13-00890-t002:** Measurement items.

Construct	Code	Item
Food healthiness	HL1	Foods in food delivery app are healthy.
	HL2	Foods in food delivery app promote my health condition.
	HL3	Foods in food delivery app are nutritional.
	HL4	Foods in food delivery app are good for health.
Eco-friendly packaging	EP1	The packaging in food delivery app is eco-friendly.
	EP2	The packaging in food delivery app is environmental.
	EP3	The packaging in food delivery app is good to protect the environment.
	EP4	The packaging in food delivery app minimizes garbage.
Trust	TR1	Food delivery app is (untrustworthy-trustworthy)
	TR2	Food delivery app is (not credible-credible)
	TR3	Food delivery app is (unreasonable-reasonable)
	TR4	Food delivery app is (dishonest-honest)
Intention to reuse	IR1	I am going to use food delivery app again.
	IR2	I will use food delivery app again.
	IR3	I will select food delivery app service for next shopping.
	IR4	I intend to use food delivery app one more time.

**Table 3 foods-13-00890-t003:** Confirmatory factor analysis results.

Construct (AVE)	Code	Mean	SD	Loading	CR
Food healthiness (0.669)	HL1	3.826	0.817	0.825	0.890
	HL2			0.836	
	HL3			0.773	
	HL4			0.835	
Eco-friendly packaging (0.713)	EP1	3.802	0.934	0.853	0.908
	EP2			0.848	
	EP3			0.876	
	EP4			0.799	
Trust (0.555)	TR1	4.126	0.649	0.780	0.833
	TR2			0.682	
	TR3			0.786	
	TR4			0.728	
Intention to reuse (0.606)	IR1	4.015	0.751	0.792	0.860
	IR2			0.819	
	IR3			0.733	
	IR4			0.768	

Note: AVE denotes average value extracted, CR stands for construct reliability, SD denotes standard deviation. χ^2^ = 420.552, df = 98, RMR = 0.038, GFI = 0.844, NFI = 0.894, RFI = 0.870, IFI = 0.917, TLI = 0.898, and CFI = 0.916.

**Table 4 foods-13-00890-t004:** Correlation matrix.

	1	2	3	4
1. Food healthiness	0.817			
2. Eco-friendly packaging	0.918 *	0.844		
3. Trust	0.662 *	0.678 *	0.744 *	
4. The intention to reuse	0.663 *	0.660 *	0.772 *	0.778

Note: Diagonal is the square root of AVE, * *p* < 0.05.

**Table 5 foods-13-00890-t005:** Results of hypothesis testing.

	Model 1 Trust		Model 2 Intention to Reuse	
	β	t Value	β	t Value
Constant	4.274	13.15 *	0.690 *	3.61
Food healthiness	−0.366	−3.16 *	0.287 *	6.53 *
Eco-friendly packaging	−0.378	−3.77 *		
Interaction	0.177	6.54 *		
Trust			0.538 *	9.70 *
F value	87.13 *		160.51 *	
R^2^	0.4354		0.4857	
Index of mediated moderation	Index	LLCI	ULCI	
	0.0956 *	0.0456	0.1703	

Note: * *p* < 0.05, interaction: healthiness × eco-friendly packaging, conditional effect of the focal predictor at values of the moderator: eco-friendly packaging (3.00): β = 0.165 *, eco-friendly packaging (4.00): β = 0.343 *, and eco-friendly packaging (4.50): β = 0.432 * (test of highest order unconditional interaction: R^2^ = 0.0713, F = 42.78 *.

## Data Availability

The data presented in this study are available on request from the corresponding author. The data are not publicly available due to privacy.

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
