# Peer review of "The Relationship between Food Healthiness, Trust, and the Intention to Reuse Food Delivery Apps: The Moderating Role of Eco-Friendly Packaging"

_foods, 2024, doi:10.3390/foods13060890_

Round 1
Reviewer 1 Report
Comments and Suggestions for Authors
1. The introduction is very unfocused and fails to clearly express the purpose of the work and how it fits into the current landscape of knowledge related to the topic at hand. Additionally, lines 67-71 redundantly reiterate that the work investigates the relationships between four key concepts: reuse, food healthiness, trust, eco-friendly packaging, without explaining, even briefly, the objectives, aims, and methods. Therefore, the introduction should be completely restructured.
2. The entire Section Two appears to resemble a review, yet the work is classified as a scientific article. Therefore, it is not appropriate to include such an extensive literature analysis, especially divided according to the four aforementioned keywords.
3. Table 6 and Figure 2 both present the same values. It would be advisable to choose one of them for inclusion in the manuscript to avoid redundancy.
4. In the conclusion section, bibliographic references should not be included.
5. In reference 5 of the bibliography, the font style needs to be standardized.
6. Certainly, including specific publications on eco-friendly packaging in the bibliography would be beneficial. E.g. https://doi.org/10.1080/14786419.2023.2235711.
7. The conclusion does not discern the achievement of the aims set in the work. Additionally, these aims were not adequately expressed in the introduction. Furthermore, the authors assert (lines 346-347) that "Although packaging is an essential issue in the food delivery app domain, its effect on consumer perception has rarely been explored." This statement is not accurate, as there is considerable recent literature on this topic (e.g.: https://doi.org/10.3389/fsufs.2020.00063).
8. The present work is in total methodological continuity with a previous work by the same authors (https://doi.org/10.3390/nu15245057) which is not cited in the bibliography.

The level of correctness and of the English language terms is quite good.
Reviewer 2 Report
Comments and Suggestions for Authors
The paper investigates the relationship between food healthiness, trust, and the intention to reuse food delivery apps, with a focus on the moderating effect of eco-friendly packaging. Using a survey methodology with 343 observations, the study employs PROCESS Model 7 for hypothesis testing. Results indicate a positive impact of trust and food healthiness on the intention to reuse, with a significant moderating effect of eco-friendly packaging on the relationship between food healthiness and trust. The study contributes to clarifying the interplay among these factors and offers managerial insights for food delivery app businesses.
My comments:
1) Can you provide more details about the sampling method used to collect data, particularly regarding the representativeness of the sample?
2) How did you ensure the reliability and validity of the survey instrument, especially considering the complex constructs involved?
3) Could you elaborate on the operationalization of variables like "trust" and "food healthiness" to better understand their measurement in the survey?
4) The paper mentions the use of PROCESS Model 7 for hypothesis testing. What specific statistical tests and procedures were employed within this framework?
5) Given the focus on eco-friendly packaging, were there any considerations for the potential influence of other packaging attributes (e.g., convenience, aesthetics) on consumer perceptions?
6) Can you discuss any potential biases or limitations associated with using Amazon Mechanical Turk as the data collection platform?
7) In the discussion section, you mentioned a potential discrepancy between the hypothesized and observed relationship between trust and healthiness. Could you provide insights into possible reasons for this discrepancy?
8) How do you envision future research expanding upon this study, particularly in terms of addressing the identified limitations?
9) Considering the managerial implications discussed, what specific strategies or interventions would you recommend to food delivery app businesses based on your findings?
Reviewer 3 Report
Comments and Suggestions for Authors
1. Abstract: the abstract does not provide sufficient conclusions results about the survey.
2. Method: there is a lack of information about the specific details of the selection procedure of the survey participants. From which area (city, rural areas, etc.) did the researchers come? How were researchers invited to participate in the survey? Moreover, the survey only contains a total of 343 observations. Does the researchers represent the American consumers?
3. Line 230, Line 232: suggest to change “Table 2” to “Table 1”.
4. Table 2: In the survey, do the participants only give a score for each attributes of the four constructs (healthiness, eco-friendly packaging, trust, and the intention to reuse)?
5. Introduction, Literature review, hypothesis development: the three sections contains most of the prior studies about the four attributes (healthiness, eco-friendly packaging, trust, and the intention to reuse). Can you explain why you write them in three parts?
Reviewer 4 Report
Comments and Suggestions for Authors
The manuscript entitled “Structural relationship between food healthiness, trust, and the intention to reuse of food delivery apps: Moderating role of eco-friendly packaging”, authored by Kyung-A Sun and Joonho Moon, presents the study dealing with the relationship among food healthiness, trust and the consumers’ intention to reuse food delivery apps. Also, the study examines the moderating effect of eco-friendly food packaging on the association between food healthiness and trust in food delivery apps. These are the main purposes of the study highlighted in the abstract and completely reflect the content of the manuscript regarding the main findings.
The Introduction: This part is outstandingly written. The author clearly pointed out the background of the study, the main purposes, and the literature review with significant number of papers cited. The only suggestion I have refers to the last paragraph. Here it should be mention how the relationships are examined. By using which methodology?
Literature review and hypothesis development: Clearly written, nicely organized, clearly defined hypotheses, nothing to add here.
Method: The authors clearly presented the research model and all the parameters used are clearly defined.
Results: The results analysis is well done. The main findings are nicely presented and interpreted. However, some minor corrections are needed. In Table 4, in the correlation matrix, please emphasize if the correlation or determination coefficients are used. I would suggest using the determination coefficients (R2) instead of r. R2 better reflects in how much % of changes in dependent variable is being influenced by the changes in the independent variable. After calculating the R2, please re-interpret the findings. The results of the hypotheses testing and the moderating effect of eco-friendly packaging are well interpreted.
Discussion: It is brief, but points out the main purpose, finding and results interpretation.
Conclusion: Well written, contains the main findings, clearly presents academic and practical implications and points out the main limitations.
Round 2
Reviewer 1 Report
Comments and Suggestions for Authors
The authors have made sure to incorporate all the suggested corrections, resulting in a higher level of specificity and completeness in the work.
Reviewer 3 Report
Comments and Suggestions for Authors
The manuscript can be accepted in present form.